# The Inflammatory Profile Correlates with COVID-19 Severity and Mortality in Cancer Patients

**DOI:** 10.3390/jpm13081235

**Published:** 2023-08-07

**Authors:** Corina Eugenia Budin, Alexandra Floriana Nemeș, Ruxandra-Mioara Râjnoveanu, Roxana Maria Nemeș, Armand Gabriel Rajnoveanu, Adrian Horațiu Sabău, Iuliu Gabriel Cocuz, Răzvan Gheorghita Mareș, Vlad Iustinian Oniga, Dariana Elena Pătrîntașu, Ovidiu Simion Cotoi

**Affiliations:** 1Pathophysiology Department, George Emil Palade University of Medicine, Pharmacy, Science and Technology of Târgu Mures, 540139 Targu Mures, Romania; cora_bud@yahoo.com (C.E.B.); sabauhoratiu@gmail.com (A.H.S.); iuliuco@gmail.com (I.G.C.); razvan_mares7@yahoo.com (R.G.M.);; 2Pneumology Department, Mures Clinical County Hospital, 540142 Targu Mures, Romania; 3Neonatology Department, Titu Maiorescu University, 040441 Bucharest, Romania; dr.alexandranemes@gmail.com; 4Palliative Medicine Department, Iuliu Hatieganu University of Medicine and Pharmacy, 400347 Cluj-Napoca, Romania; 5Faculty of Medicine, Titu Maiorescu University, 67A Gheorghe Petrascu Str., 031593 Bucharest, Romania; roxanamarianemes@gmail.com; 6Occupational Medicine Department, Iuliu Hatieganu University of Medicine and Pharmacy, 400347 Cluj-Napoca, Romania; armand.rajnoveanu@umfcluj.ro; 7Pathology Department, Mures Clinical County Hospital, 540142 Targu Mures, Romania; 8Faculty of Medicine, George Emil Palade University of Medicine, Pharmacy, Science and Technology of Târgu Mures, 540139 Targu Mures, Romania; vladiustinian@yahoo.com

**Keywords:** lung cancer, COVID-19 severity, inflammatory profile, COVID-19 mortality

## Abstract

Background: The correlation of the inflammatory profile with the severity of the disease in neoplastic patients with SARS-CoV-2 infection was addressed. Methods: A database of 1537 patients hospitalized in the pneumology department was analyzed. After applying the inclusion and exclusion criteria, 83 patients (67% males, 33% females) were included. Results: Most of the analyzed patients were hospitalized with a moderate form of disease, explaining the significant percentage of 25% mortality. The frequency of the type of neoplasm was higher for lung cancer, followed by malignant colon tumor. We identified a significant association between the increased value of ferritin (*p* < 0.0001, OR = 22.31), fibrinogen (*p* = 0.009, OR = 13.41), and C-reactive protein (*p* = 0.01, OR = 7.65), respectively, and the level of severity of COVID-19. The results of the univariate logistic regression analysis for predicting the severity of the disease revealed that the increased values of ferritin (*p* = 0.001, OR = 22.31) and fibrinogen (*p* = 0.02, OR = 13.41) represent a risk for a serious negative prognosis of COVID-19. Conclusions: Our study demonstrated that the value of the analyzed inflammatory parameters increased in direct proportion to the severity of the disease and that higher values were associated with increased mortality in the study group.

## 1. Introduction

Infection with severe acute respiratory syndrome coronavirus 2 (SARS-CoV-2), the cause of coronavirus disease 2019 (COVID-19), led to a pandemic that had global effects on all human activities. The clinical presentation of severe COVID-19 includes a broad spectrum of clinical illness, particularly acute respiratory distress syndrome, cytokine release syndrome, multiorgan failure, and death [1]. Direct viral injuries and uncontrolled inflammation have been suggested as contributing factors to the severity of COVID-19. The COVID-19 pandemic highlighted the critical role of the host’s effective immune response in controlling the viral infection and demonstrated the devastating effect of immune dysregulation [1,2,3]. Due to the critical role of inflammation in the pathogenesis and progression of COVID-19, as well as the increase of pro-inflammatory cytokines in the serum of patients with COVID-19, many attempts were made to investigate the role of inflammatory markers in the predictive evolution of COVID-19 infection [4,5]. Higher levels of inflammatory markers, including C-reactive protein (CRP) and the neutrophil-to-lymphocyte ratio, as well as various inflammatory cytokines and chemokines, have been shown to be related to a more severe clinical course in patients with COVID-19 [4,6].

Cancer patients are more vulnerable to SARS-CoV-2 infection, given the relatively high prevalence of underlying and cancer-induced chronic diseases and systemic immunosuppressive chemotherapy. SARS-CoV-2-infected cancer patients may be at higher risk of severe events and may deteriorate more rapidly than patients without this viral condition [7,8,9]. The prediction of prognosis for patients with COVID-19 and cancer can help prioritize patient-care resources and inform patients about the choice of treatment strategy, which would likely improve patient prognosis [10,11].

From the category of neoplastic patients, lung-cancer patients have the highest risk of becoming infected with COVID-19 [12]. In addition, the mortality of patients with cancer and concomitant SARS-CoV-2 infection is higher than it is in the general population. These results may be due to interconnectivity between respiratory symptoms from COVID-19 and respiratory symptoms of lung-cancer patients [13,14]. Tumor cells are more susceptible to viral replication and, ultimately, result in decreased antiviral immunity [14,15]. Another hypothesis stated in the literature is that inflammatory changes triggered by COVID-19 infection can determine cell transformation and, implicitly, trigger the pathophysiological cascade of neoplastic transformation [16,17].

The aim of our study was to establish the degree of correlation between the parameters of the inflammatory profile (ferritin, fibrinogen, leukocyte count, and CRP) and the degree of severity of COVID-19 in patients with a concomitant diagnosis of neoplasm.

## 2. Materials and Methods

We conducted a retrospective study, collecting data from the medical records of patients diagnosed with neoplasm and infected with the SARS-CoV-2 virus. These patients were admitted to the pneumology department of the Mureș County Clinical Hospital between 1 January 2020 and 31 March 2022. Admission to the pulmonology department was carried out directly from the emergency unit—the patients had not previously been hospitalized in other departments of our hospital. The study was conducted in accordance with the European and National legislation directives and with the principles stated in the Helsinki declaration. It received approval number 16944/09.11.2022 from the Ethics Committee of the Clinical County Hospital Mureș, Romania.

The information collected was as follows: gender, age, main diagnosis, secondary diagnoses, comorbidities, survival status, laboratory analyses, and inflammatory markers, including the level of ferritin, the level of fibrinogen, the total number of leukocytes, and the value of C-reactive protein. The values of inflammatory parameters that were analyzed in our study were the values at admission. Therefore, the length of hospitalization was not analyzed in relation to these values.

### 2.1. Inclusion Criteria

Patients (aged over 18 years) continuously hospitalized in the pneumology department with a diagnosis of SARS-CoV-2 infection, confirmed by a real-time polymerase chain reaction (RT-PCR) test, and a concomitant diagnosis of neoplasia.

### 2.2. Exclusion Criteria

Continuously hospitalized patients with a SARS-CoV-2 infection, confirmed RT-PCR, but without a concomitant neoplasm diagnosis.Continuously hospitalized patients with SARS-CoV-2 infection, confirmed RT-PCR, and a concomitant neoplasm diagnosis, but for whom data on the inflammatory profile were not available.

According to the Official Monitor of Romania, no. 978, published on 13 October 2021, the classification of the forms of COVID-19 was made as follows [18]:Mild form: patients with general symptoms and/or upper respiratory tract symptoms, without evocative manifestations of pneumonia and without lung damage;Medium form: patients with imaging-confirmed pneumonia, but without hypoxemia (if there was no respiratory damage prior to the current disease);Severe form: patients with respiratory distress, with SaO_2_ below 94% in atmospheric air, and imaging abnormalities of lung damage;Critical form: patients with severe respiratory failure with the need for ventilatory support, with septic shock and/or multiple organ failure [16].

For the first part of the analysis, the patients were divided into three groups, according to the severity of COVID-19: Group 1—mild form; Group 2—medium form; and Group 3—severe form. For the second part of the analysis, we divided the patients into two groups according to survival status: Group 4—surviving patients; Group 5—deceased patients.

### 2.3. Statistical Processing

Statistical processing was performed using the statistical software GraphPad Prism V9.01 for Windows and the SPSS V20 package (IBM). Statistical analysis involved applying tests for paired and unpaired data, with the final study group consisting of 83 subjects. Calculated descriptive statistics indicators included skewness and kurtosis of the distribution curve, minimum value, maximum value, standard deviation, mean, standard error, mode, median, and variance. Continuous variables were expressed as mean ± standard deviation (SD). Categorical variables were presented as number and percentage. To test the normality of continuous quantitative data, we used the Kolmogorov–Smirnov (KS) test. When the results of the KS test indicated that the data for the variables in one studied group had a parametric distribution and the data for the variables in the other studied group had a non-parametric distribution, we chose to continue applying significance tests for data with non-Gaussian distribution—namely, the Wilcoxon test and the Mann–Whitney test—and when the results of the KS test indicated that both groups had a parametric distribution, we chose to use the Student’s *t*-test for dependent data. For the analysis of categorical data, we used the chi-square significance test or the Fischer test. Significance tests (Mann–Whitney and Student’s *t*-test) were used to evaluate whether there were significant differences between the medians/means of different laboratory parameters, and we used ANOVA analysis when we analyzed more than two groups. We used the receiver operating characteristic (ROC) analysis to determine the cut-off values for the inflammatory status parameters. In addition, we used univariate and multivariate logistic regression to determine the predictive factors. We also applied the Spearman correlation analysis to determine the relational degree between certain parameters. The significance threshold chosen was alpha = 0.05, and *p* was considered significant when *p* ≤ alpha.

To assess the correlation between the analyzed inflammatory parameters, we used the Pearson correlation. The correlation coefficient had a value between −1 and 1. If the correlation coefficient was r = 0, then it meant that we had no correlation between the two variables. If the correlation coefficient was r = +1, then we had a perfect correlation, and if r = −1, we had a perfect inverse correlation. In order to classify the intensity of the correlation between the independent and the dependent variables, we considered the following ranges in absolute value: very weak association (r between 0 and 0.19), weak association (r between 0.20 and 0.39), moderate association (r between 0.40–0.59), strong association (r between 0.60 and 0.79), and very strong association (r between 0.8 and 1).

## 3. Results

A database of 1537 patients hospitalized between 1 January 2020 and 31 March 2022 in the pneumology department of the Mureș County Clinical Hospital was analyzed. Following the analysis of the inclusion and exclusion criteria, the analyzed group included 83 patients with a concurrent diagnosis of COVID-19 and neoplasm. Among them, the percentage of males was higher than females (67% males and33% females). In the study group, the frequency of the type of neoplasm was highest for lung cancer (*n* = 23), followed by malignant colon tumors (*n* = 17), malignant tumors of the genitourinary system (*n* = 9), malignant breast tumors (*n* = 7), prostate tumors (*n* = 6), and—in descending order of frequency—other tumors of the digestive system, unspecified malignant tumors, and malignant tumors of the ovary, tongue, abdomen, pelvis, brain, larynx, peritoneum, thyroid, lymphatic system (Table 1).

Among associated comorbidities, the highest frequency was recorded for cardiovascular pathology (*n* = 22), followed—in descending order—by diabetes (*n* = 10), obesity (*n* = 8), liver diseases (*n* = 5), and neurological diseases (*n* = 3).

Descriptive statistical indicators of laboratory parameters are summarized in Table 2. Age was analyzed, as well as the levels of ferritin, fibrinogen, CRP, and leukocyte count. For each of the parameters, the mean value and the median value, as well as the standard deviation, were calculated. The minimum value and the maximum value of each analyzed parameter were analyzed to ensure the most faithful interpretation of the average values obtained.

Following the results of the ROC analysis for the inflammatory parameters, the following cut-off values were established: for ferritin, the value of 425 ng/mL; for fibrinogen, the value of 472 mg/dL; for CRP, the value of 8 mg/dL; and for leukocytes, the value of 4780/mmc. The distribution of the inflammatory parameters on the ROC curve is illustrated in Figure 1.

Applying the correlation analysis between the laboratory parameters, significant positive correlations of moderate intensity were obtained among ferritin, fibrinogen, and CRP respectively, and significant positive correlations of weak intensity was found between fibrinogen and CRP (Table 3).

### 3.1. Severity Scores

Descriptive statistical parameters were also correlated with disease severity (Table 4). Average ferritin values, fibrinogen values, CRP values, and white blood cell counts were higher in severe forms of the disease, compared to mild and moderate forms.

Regarding the degree of severity, most of the analyzed patients were hospitalized with a moderate form of COVID-19, which explains the significant percentage of mortality: 25%, *N* = 21. The number of patients distributed according to the severity of COVID-19 is illustrated by Figure 2a: *n* = 40 patients presented with the medium form of the disease, *n* = 27 patients presented with the severe form of the disease, and *n* = 16 patients presented with the mild form of the disease.

### 3.2. Disease Severity

After analyzing the data, we did not obtain significant differences, *p* = 0.16, between the average ages of the three groups of patients, depending on the severity of COVID-19 (Figure 2b). We highlight the fact that patients with the severe form of COVID-19 had a higher average age than patients with the medium and mild forms (72.44 years versus 67.92 years and 69.93 years, respectively).

In light of the study’s aims, the minimum, maximum, and average values of the analyzed parameters were centralized, correlating these values with the severity of the disease (Table 5).

The results of the univariate logistic regression analysis for predicting the severity of the disease revealed that the increased values of ferritin (*p* = 0.001, OR = 22.31 [4.57–108.81]) and fibrinogen (*p* = 0.02, OR = 13.41) [1.49–120.18] represented a risk for a serious negative prognosis of COVID-19 (Table 6). Similar values were obtained following the multivariate analysis (Table 7).

We identified a significant association between the increased value of ferritin (*p* < 0.0001, OR = 22.31) (Figure 3), fibrinogen (*p* = 0.009, OR = 13.41) (Figure 4), and C-reactive protein (*p* = 0.01, OR = 7.65) (Figure 5) and the level of severity of COVID-19. Patients with increased values of these parameters presented with a severe form of COVID-19 (Table 8).

Regarding the degree of relationship between the parameters of the inflammatory status and the level of severity of the disease, we identified a moderate positive correlation between ferritin and the degree of severity of the disease (*p* = 0.0001) and weak positive correlations between fibrinogen (*p* = 0.005) and CRP (*p* = 0.004) and disease severity. We did not obtain a significant correlation between leukocytes and disease severity (*p* = 0.11) (Table 9).

We observed a significant difference between the average values of ferritin in the three groups of patients studied—namely, the value of ferritin increased significantly from the mild form to the moderate form and the value of ferritin increased significantly from the moderate form to the severe form, *p* = 0.01 (Figure 6a). We also obtained a significant difference between the average values of the leukocytes in the three groups of patients studied—namely, the value of leukocytes increased significantly from the mild form to the moderate form and the value of leukocytes increased significantly from the moderate form to the severe form, *p* = 0.02. For CRP, a difference was observed between the mild form (mean value 26.9%), the moderate form (mean value 69.52), and the severe form (mean value 77.52%), but without reaching the threshold of statistical significance (*p* = 0.10).

To present the results according to survival status, we divided the patients into two groups: Group 4—survivor patients and Group 5—deceased patients.

Descriptive statistics were reported according to survival (Table 10). We analyzed the same parameters (age, minimum value, maximum value, and mean value for ferritin, fibrinogen, CRP, and leukocyte count) for both categories of hospitalized patients, survivors and deceased.

Of the total of 83 analyzed patients, 21 died. The results showed that the patients who died were older (mean age of 72.76 years in those who died versus mean age of 68.72 years in those who survived), but the age difference was not significant (*p* = 0.14). In the group of deceased patients, the ferritin values were significantly higher than those in living patients (*p* = 0.004) (Figure 6b). The value of fibrinogen in the group of deceased patients (574.05 mg/dL) was higher than it was in the group of living patients (516.27 mg/dL), but without meeting the threshold of statistical significance (*p* = 0.12). The value of leukocytes in the group of deceased patients was significantly increased (10,479.5/mmc) compared to the group of living patients (7563.71/mmc), *p* = 0.04. The difference between the CRP value in the group of deceased patients and the group of living patients did not reach the threshold of statistical significance (*p* = 0.72) (66.69 mg/dL vs. 65.70 mg/dL).

## 4. Discussion

The correlation of inflammatory biomarker values with COVID-19 severity is known from the literature. One of the aims of our study was to demonstrate that this correlation is also maintained in neoplastic patients with a concomitant SARS-CoV-2 infection.

One of the key features of COVID-19 is the exacerbated inflammatory status seen in some patients, especially those who develop severe disease [19,20]. Specialized studies have demonstrated that several types of immune cells and inflammatory mediators are involved in the process and progression of the disease [21,22].

In the present study, the correlation of the inflammatory profile with the severity of the disease in neoplastic patients with SARS-CoV-2 infection was addressed. The experience accumulated during the conduct of this study demonstrated that the value of the analyzed inflammatory parameters increased in direct proportion to the severity of the disease, so that the highest values were recorded in severe forms of the disease, compared to mild and moderate forms of the disease. At the same time, very high values of these parameters were associated with increased mortality in the study group.

A study by Gong et al., on a batch of 100 patients with mild, moderate, and severe forms of COVID-19 infection, supported the hypothesis that the mortality rate increases significantly with the clinical worsening of the patients and with the progression of the disease to severe forms. At the same time, severe forms of the disease were associated with an increase in inflammatory markers (ferritin, ferroprotein, CRP, IL8, IL 17, and TNFα) [23,24]. These results were confirmed in the specialized literature, both nationally and globally, which certified the validity of the data obtained in our study [6,25].

C-reactive protein (CRP) is a nonspecific acute-phase reactant that has elevated levels when a patient has infection or inflammation. Higher levels indicate a more severe infection, and such levels have been used as markers of the severity of COVID-19. Elevated levels of serum C-reactive protein have been observed in patients with COVID-19 and have been used to aid in the following triage: diagnosis of infection, quantification of inflammatory status, and prognosis of the patients [26,27]. Crucially, the levels can increase before a patient’s vital signs are affected or before the leukocytes are increased [6,27].

The data obtained in our study showed increases of leukocytes, CRP, fibrinogen, and ferritin in direct proportion to the degree of severity of the disease. Regarding mortality, increased levels of ferritin and total leukocyte count were obtained (*p* = 0.04). For fibrinogen and CRP, the values in deceased patients were higher than they were in the group of surviving patients, but without statistical significance (*p* = 0.12, respectively 0.72).

Similar results were highlighted by a study conducted on a group of 283 patients with RT-PCR testing that confirmed COVID-19, in which three inflammatory markers—serum CRP (86.36% sensitivity; 88.89% specificity), lactate dehydrogenase (LDH) (90.91% sensitivity; 80.56% specificity), and ferritin (95.45% sensitivity; 86.57% specificity)—were shown to be useful in predicting mortality [28]. Zeng et al. showed that a number of inflammatory markers of high sensitivity, such as serum soluble IL-2 receptor, IL-6, IL-8, IL-10, TNF-α, procalcitonin, ferritin, LDH, and CRP, together with the lymphocyte high-sensitivity PCR ratio, had very high levels in critically ill COVID-19 patients [29].

In our study of the evaluated group, the analysis identified a cut-off value for CRP of 8 ng/mL. Thus, patients with a CRP value above this threshold had to be monitored more closely and were at risk of developing a severe form of the disease. The differences in the CRP values between patients with a mild form of the disease and those with moderate and severe forms of the disease were significant, but the difference between patients with moderate and severe forms of the disease did not reach the threshold of statistical significance (*p* = 0.10). This result is partially consistent with the conclusion of another study conducted on 100 patients, in which it was observed that a CRP value > 6.2 ng/mL indicated the progression of the disease to a severe form [23].

Ferritin is a key mediator of immune dysregulation, particularly in extreme hyperferritinemia, through direct immunosuppressive and proinflammatory effects, contributing to the cytokine storm [30]. Research [31,32] indicated that fatal outcomes of COVID-19 were accompanied by cytokine-storm syndrome; therefore, it has been suggested that the severity of the disease depends on cytokine-storm syndrome. Many people with diabetes have elevated serum ferritin levels [32,33] and are known to be more likely to experience serious complications from COVID-19 [32,34].

In a study that included 20 patients with COVID-19, it was found that people with severe and very-severe COVID-19 had an increased level of serum ferritin [33]. The serum-ferritin level in the very-severe COVID-19 group was significantly higher than it was in the severe COVID-19 group (1006.16 ng/mL [IQR: 408.265–1988.25] vs. 291.13 ng/mL [IQR: 102.1–648.42], respectively) [35].

Another study demonstrated, similarly, that in patients who died of COVID-19, ferritin levels were elevated at hospital admission and throughout hospitalization [31]. In addition, Chen et al. [36] analyzed the clinical characteristics of 99 patients, 63 of whom had serum ferritin well above the normal range. Increased ferritin values were also found in the autopsies of 12 patients whose cause of death was SARS-CoV-2 infection. An analysis of the peripheral blood of 69 patients with severe COVID-19 revealed elevated levels of ferritin, compared to patients with non-severe disease. All of these results of the above-mentioned studies were consistent with the results of our study, where the value of serum correlated with statistical significance both with the severity of the disease (*p* < 0.0001, OR = 22.31) and with mortality (*p* = 0.004).

Coagulopathy in COVID-19 differs from usual disseminated intravascular coagulation by having elevated fibrinogen, normal or slightly prolonged prothrombin time, and activated partial thromboplastin time, with a platelet count > 100 × 10^3^/mL, but no significant bleeding [37].

Elevated D-dimer levels are seen very frequently in patients with COVID-19. Several meta-analyses have shown that D-dimer levels have prognostic value and correlate with disease severity and in-hospital mortality [29,38,39]. A level > 2.0 μg/mL on admission could be a useful predictor of mortality. D-dimer may be an early marker to guide the management of patients with COVID-19 [29].

Changes in hemostasis, with a tendency toward a hypercoagulable state, have been reported in patients with COVID-19. Elevated levels of D-dimer and fibrinogen were associated with poor prognosis of hospitalized patients. Fibrinogen levels greater than 617 mg/dL are more likely to be associated with a severe clinical form of the disease, characterized by acute respiratory distress syndrome on admission, and with increased risk of thromboembolic events during hospitalization, particularly pulmonary embolism and embolism-related pulmonary death [40]. These results were supported in our research, where 69.10% of the patients had fibrinogen values above 472 mg/dL. The inflammatory response in neoplastic patients plays the primary role in the defense reaction to infectious agents and mediates the reparative reaction and postinfectious tissue remodeling [7,8]. Among the type of neoplasia in our study group, the largest number of patients presented with malignant lung tumors (*n* = 23), followed by malignant tumors of the colon (*n* = 17) and malignant tumors of the breast (*n* = 7). This association can be explained by the important inflammatory changes at lung level that are present both in the bronchopulmonary neoplasm and in the respiratory damage from the medium and severe forms of SARS-CoV-2 infection. These three localizations of neoplasia, in association with SARS-CoV-2 infection, can also be found in the specialized literature. In the study by Li et al., the most common location was breast cancer, followed by colorectal cancer and lung cancer [1,41]. The fatality rate in the study group was lower than it was in the literature (25% vs. over 30%) [14].

Of the mild-form COVID-19 patients included in the study, the majority (*n* = 8) were diagnosed with colon cancer, followed by those with bronchopulmonary cancer *(n* = 3). The predominance of patients with colon cancer was also recorded in patients with severe forms of disease (*n* = 9), followed by, as an absolute value, patients with lung cancer (*n* = 7).

The increased frequency of colon cancer in patients included in the study may be due to the fact that colon neoplasm is on the third position of incidence in men and the second position of incidence in women [13,42]. On the other hand, colon-cancer patients are of older ages and have associated comorbidities, which could explain the high frequency in those patients with severe forms of the disease [42].

The limitations of the study were, on the one hand, the relatively small group of patients and, on the other hand, the fact that only the numbers of leukocytes, fibrinogen, ferritin, and CRP among the inflammatory markers were analyzed. In these patients, there were no resources to analyze other parameters, such as IL6 or IL8. In addition, due to the fact that the number of patients in whom procalcitonin was collected was reduced, this parameter was excluded from the interpretation of the results, although some of the severe and critical cases had symptoms of sepsis. The values of inflammatory markers were not available in these patients in the pre-pandemic and post-pandemic periods. These values are available only for lung-cancer patients, who are chronic patients of the pulmonology department. In the context of neoplastic pathology, the value of these biomarkers is constantly maintained at the upper limit of normal or at elevated values, but these values are much lower than those recorded when COVID-19 was concomitant with neoplastic pathology.

Another limitation of our study is that we did not assess the impact of the neoplasia stage on the course of COVID-19. However, the patients admitted to the pulmonology department and included in this study were patients receiving ongoing oncological treatment (mostly chemotherapy).

Although COVID-19 infection in neoplastic patients represents a cumulation of fatality, the evolution of these patients was not necessarily severe. One of the explanations is that these patients were much more closely monitored than patients with other chronic pathologies, and they were tested before each chemotherapy cycle. The mortality of 25% registered in the study group was lower than expected, which could demonstrate that careful follow-up of these patients and non-interruption of oncological treatment can contribute to a slightly more favorable prognosis for these patients. The global analysis of these patients—without being differentiated by type of neoplastic pathology—leads to the conclusion that the evolution to severe forms of COVID-19 is determined by the magnitude of the inflammatory profile and not by the associated comorbidities.

The strength of this study was not only evaluating the impact of COVID-19 infection on patients diagnosed with certain types of neoplasia, but also bringing toto the attention of clinicians the correlation between the most accessible inflammatory biomarkers and the prognosis of these patients, thus leading to an accessible panel of investigations that can guide doctors by optimizing their therapeutic behavior and their medical resources.

## 5. Conclusions

Our study demonstrated that the simultaneous presence of malignant tumors and SARS-CoV-2 infection represents an unfavorable prognostic factor in the progress of the patient. An increased value of inflammatory parameters was associated with both increased severity and increased mortality. Among these parameters, the value of ferritin was statistically significantly correlated with the severity of the disease and with mortality; the number of leukocytes was correlated with the severity of the disease; the value of fibrinogen was higher in patients with severe forms of the disease, but without reaching the threshold of statistical significance; and the CRP value had increased values in patients with moderate and severe forms of the disease.

Each patient with SARS-CoV-2 infection, regardless of the form of severity, must be individually approached, taking into account the associated comorbidities. The outcomes are unpredictable, and the evaluation of the inflammatory profile can be an important part of the complex approach for each patient.

## Figures and Tables

**Figure 1 jpm-13-01235-f001:**
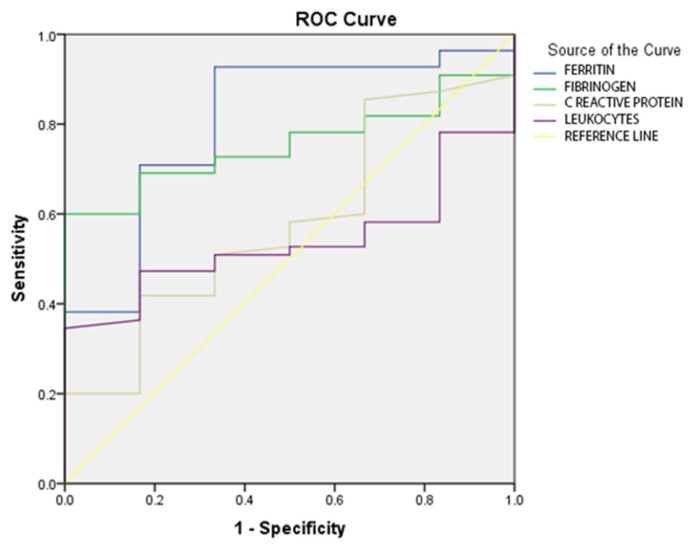
ROC curve for inflammatory-status parameters in the study group.

**Figure 2 jpm-13-01235-f002:**
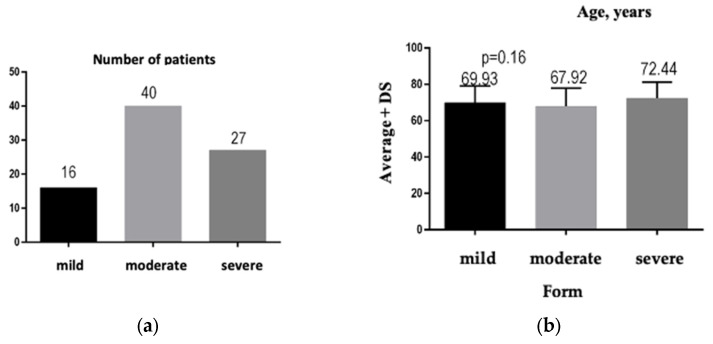
(**a**) Graphical representation of the number of patients according to the disease severity. (**b**) Graphical representation of the average age of patients for each form of COVID-19.

**Figure 3 jpm-13-01235-f003:**
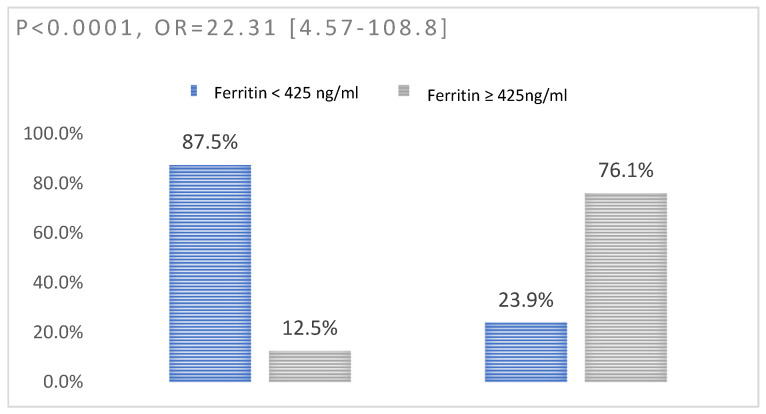
Association between ferritin and disease severity: left—non-severe disease; right—severe disease.

**Figure 4 jpm-13-01235-f004:**
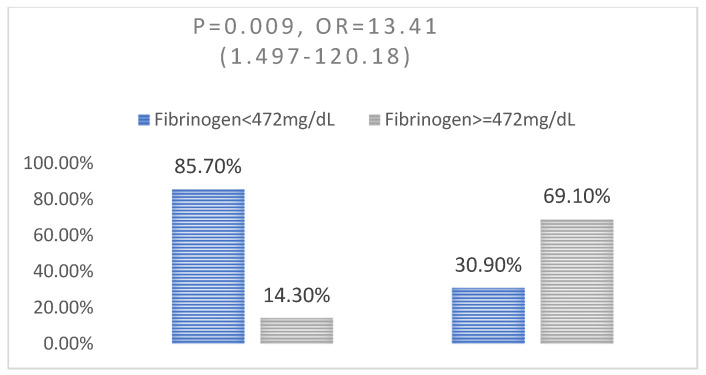
Association between fibrinogen and disease severity: (left—non-severe disease; right—severe disease.

**Figure 5 jpm-13-01235-f005:**
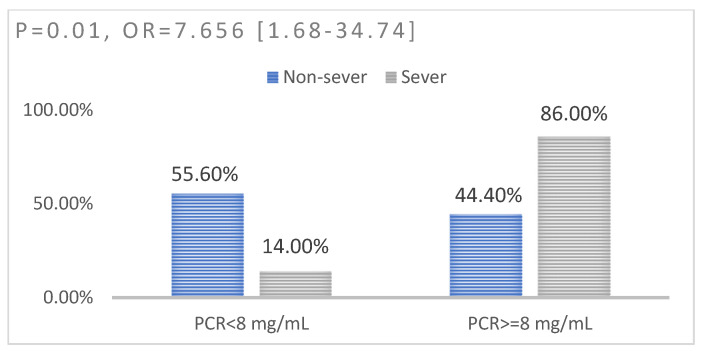
Association between CRP and disease severity: left—non-severe disease; right—severe disease.

**Figure 6 jpm-13-01235-f006:**
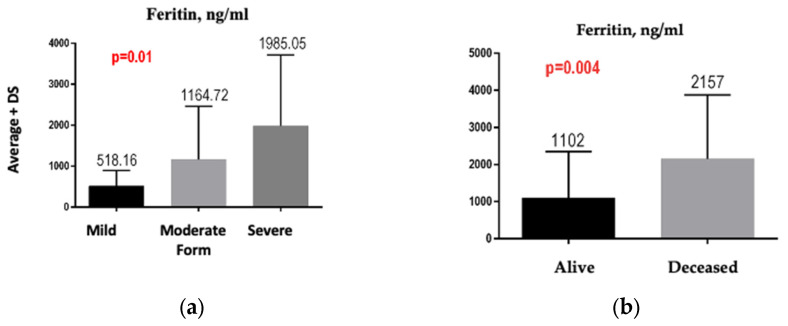
(**a**) Comparison between ferritin values according to the severity of COVID-19. (**b**) Ferritin value, depending on survival.

**Table 1 jpm-13-01235-t001:** Frequency of the types of neoplasm presented by the patients of the study.

Tumor Type	Frequency	Percent	Valid Percent	Cumulative Percent
Malignant neoplasm of abdomen	1	1.2	1.2	1.2
Malignant neoplasm of the digestive system	4	4.8	4.8	6.0
Malignant neoplasm of the reproductive system	2	2.4	2.4	8.4
Malignant neoplasm of the urogenital system	9	10.8	10.8	19.3
Malignant neoplasm of the pelvis	1	1.2	1.2	20.5
Malignant bronchopulmonary neoplasm	23	26.5	26.5	47.0
Malignant neoplasm of the colon	17	20.5	20.5	67.5
Malignant brain neoplasm	1	1.2	1.2	68.7
Malignant neoplasm of the larynx	1	1.2	1.2	69.9
Malignant neoplasm of the tongue	2	2.4	2.4	72.3
Unspecified malignant neoplasm	3	3.6	3.6	75.9
Malignant ovarian neoplasm	3	3.6	3.6	79.5
Peritoneal malignant neoplasm	1	1.2	1.2	80.7
Malignant prostate neoplasm	6	7.2	7.2	88.0
Malignant breast neoplasm	7	8.4	8.4	96.4
Lymphatic system malignant neoplasm	1	1.2	1.2	97.6
Malignant thyroid neoplasm	1	1.2	1.2	98.8
Total	83	100.0	100.0	

**Table 2 jpm-13-01235-t002:** Descriptive statistical indicators of the study group.

		Age	Ferritin	Fibrinogen	C-Reactive Protein	Leukocyte
N	Valid	83	61	62	66	83
	Missing	0	26	25	21	0
Mean ± SD		69.75 ± 9.804	1431.28 ± 1486.083	534.00 ± 174.324	66.00 ± 75.144	8301.45 ± 5883.042
Median		72.00	840.00	497.50	40.00	7620.00
Minimum		45	159	288	0	0
Maximum		89	7500	999	320	31,790
Sum		5789	87,308	33,108	4356	689,020

SD—standard deviation.

**Table 3 jpm-13-01235-t003:** Correlation analysis between laboratory parameters.

		Age	Ferritin	Fibrinogen	C-Reactive Protein	Leukocyte
Age	Pearson correlation	1	−0.161	−0.028	−0.222	0.016
	Sig. (2-tailed)		0.215	0.831	0.073	0.884
	N	83	61	62	66	83
Ferritin	Pearson correlation	−0.161	1	0.557	0.403	0.146
	Sig. (2-tailed)	0.215		0.000	0.001	0.260
	N	61	61	61	61	61
Fibrinogen	Pearson correlation	−0.028	0.557	1	0.355	0.131
	Sig. (2-tailed)	0.831	0.000		0.005	0.310
	N	62	61	62	62	62
C-reactive protein	Pearson correlation	−0.222	0.403	0.355	1	0.011
	Sig. (2-tailed)	0.073	0.001	0.005		0.932
	N	66	61	62	66	66
Leukocyte	Pearson correlation	0.016	0.146	0.131	0.011	1
	Sig. (2-tailed)	0.884	0.260	0.310	0.932	
	N	83	61	62	66	83

**Table 4 jpm-13-01235-t004:** Correlation of statistical parameters with the degree of severity of COVID-19.

Parameter/Form of the Disease		*N*	Mean ± SD	Std. Error	95% Confidence Interval for Mean	95% Confidence Interval for Mean	Minimum	Maximum
					Lower Bound	Upper Bound		
Age	mild	16	69.94 ± 9.19	2.299	65.04	74.84	46	79
	moderate	40	67.85 ± 10.46	1.654	64.50	71.20	45	89
	severe	27	72.44 ± 8.78	1.690	68.97	75.92	48	87
Ferritin	mild	6	518.00 ± 8.78	154.834	119.99	916.01	179	1220
	moderate	32	1204.50 ± 1295.78	229.064	737.32	1671.68	159	7500
	severe	23	1985.04 ± 1732.49	361.250	1235.86	2734.23	166	5353
Fibrinogen	mild	7	405.00 ± 69.52	26.279	340.70	469.30	290	483
	moderate	32	520.03 ± 163.06	28.826	461.24	578.82	295	899
	severe	23	592.70 ± 190.5	39.724	510.31	675.08	288	999
CRP	mild	9	26.78 ± 33.76	11.256	0.82	52.73	0	102
	moderate	34	68.65 ± 75.72	12.987	42.23	95.07	1	263
	severe	23	77.43 ± 83	17.308	41.54	113.33	2	320
WBC count	mild	16	5.26 ± 3.971	992.831	3143.83	7376.17	0	10,750
	moderate	40	8.234 ± 4.346	687.169	6844.57	9624.43	0	23,750
	severe	27	10.202 ± 7.891	1518.648	7081.34	13,324.59	0	31,790

Data presented as mean ± standard deviation. WCC—white cell count (× 10^3^/mcl); CRP—C-reactive protein (mg/L); ferritin (mcg/L); SD—standard deviation.

**Table 5 jpm-13-01235-t005:** The minimum, maximum, and average values for the investigated parameters, according to the severity of the disease.

	Ferritin, ng/mL			CRP, mg/mL			Fibrinogen, mg/dL			WBC, U/µL × 10^3^		
Form	Min	Max	Average	Min	Max	Average	Min	Max	Average	Min	Max	Average
Mild	179	1220	518	0	102	26.78	290	483	405	0	10.750	5.260
Moderate	159	7500	1204.5	1	263	68.65	295	899	520.03	0	23.750	8.234
Severe	166	5353	1985.04	2	320	77.43	288	999	592.7	0	31.790	10.202

**Table 6 jpm-13-01235-t006:** Univariate logistic regression for predicting the severity of COVID-19.

	B	S.E.	Wald	df	Sig.	Exp(B)	95% C.I. for EXP(B)	
							Lower	Upper
Ferritin	3.105	0.808	14.753	1	0.001	22.312	4.575	108.811
Fibrinogen	2.596	1.119	5.384	1	0.02	13.412	1.497	120.184
CRP	0.018	0.012	2.358	1	0.125	1.01	0.995	1.041
Leukocyte	0.908	0.58	2.453	1	0.117	2.479	0.796	7.72

**Table 7 jpm-13-01235-t007:** Multivariate logistic regression for predicting the severity of COVID-19.

	B	S.E.	Wald	df	Sig.	Exp(B)	95% C.I. for EXP(B)	
							Lower	Upper
Ferritin	3.383	1.242	7.425	1	0.006	29.469	2.585	335.921
Fibrinogen	2.468	1.393	3.141	1	0.076	11.804	0.770	180.984
CRP	−0.371	1.432	0.067	1	0.796	0.690	0.042	11.420
Leukocyte	−0.776	1.388	0.312	1	0.576	0.460	0.030	6.999

**Table 8 jpm-13-01235-t008:** Association between inflammatory biomarkers and disease severity.

	OR	95% Confidence Interval		X^2^	*p* Value
Ferritin	22.313	4.575	108.811	22.64	<0.0001
Fibrinogen	13.412	1.497	120.184	7.99	0.009
CRP	7.656	1.687	34.74	8.47	0.01
WBC count	2.479	0.796	7.723	2.54	0.12

**Table 9 jpm-13-01235-t009:** Spearman correlation analysis between disease severity and inflammatory parameters.

	Disease Severity	
Parameter	r(ρ)	*p* value
Ferritin	0.52	0.0001
Fibrinogen	0.36	0.005
CRP	0.35	0.004
Leukocyte	0.17	0.11

**Table 10 jpm-13-01235-t010:** Descriptive statistical indicators of the parameters, according to survival.

		*N*	Mean	95% Confidence Interval for Mean		Minimum	Maximum
				Lower Bound	Upper Bound		
Age	Alive	62	68.73 ± 10.304	66.11	71.34	45	89
	Deceased	21	72.7 ± 7.582	69.31	76.21	54	87
Ferritin	Alive	42	1102.95 ± 1253.203	712.43	1493.48	159	7500
	Deceased	19	2157.05 ± 1725.295	1325.49	2988.62	166	5353
Fibrinogen	Alive	43	516.28 ± 172.608	463.16	569.40	290	999
	Deceased	19	574.11 ± 176.119	489.22	658.99	288	900
CRP	Alive	47	65.74 ± 78.129	2.81	88.68	0	320
	Deceased	19	66.63 ± 69.213	33.27	99.99	2	230
WBC count	Alive	62	7.563 ± 4.608	6.393	8.733	0	23.750
	Deceased	21	10.479 ± 8.394	6.658	14.300	0	31.790

Data presented as mean ± standard deviation. WCC—white cell count (×10^3^/mcl); CRP—C-reactive protein (mg/L); ferritin (mcg/L).

## Data Availability

Not applicable.

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
