# Peer review of "The Inflammatory Profile Correlates with COVID-19 Severity and Mortality in Cancer Patients"

_jpm, 2023, doi:10.3390/jpm13081235_

Round 1
Reviewer 1 Report
An interesting study for the association of the inflammatory profile with disease severity focusing on cancer patients with SARS Cov2 infection. The study involves 83 hospitalized patients with cancer and CIVID-19.
The study revealed that increased values of inflammatory markers were associated with more severe forms of the disease.
A few comments
1. In the abstract: PCR value… ??? do the authors mean a measure of viral load? please clarify.
2. In the ethics section there is no mention of informed consent form, was this waived by the ethics committee? please clarify.
3. In the inclusion criteria is mentioned “continuously hospitalized” please explain, preferably, provide a time limit for this.
4. In the statistics section: “Calculated descriptive statistics indicators include: skewness and kurtosis of the distribution curve, minimum value, maximum value, standard deviation, mean, standard error, mode, median, and variance.” Not all these statistics are presented, please report only the statistics relevant to the results of the manuscript.
5. Table 2, three decimal points are too much, propose to use 2 decimals, similarly for all SDs.
6. Figure 1, in the ROC analysis is not clear the golden standard. Please mention this in the figure caption and in the relevant paragraph.
7. This sentence fits to the materials and methods, statistics: “To assess the correlation between the analysed inflammatory parameters, we used 179 the Pearson correlation. The correlation coefficient can have a value between -1 and 1. If 180 the correlation coefficient is r=0, then it means that we have no correlation between the 181 two variables. If the correlation coefficient is r=+1, then we have a perfect correlation, and 182 if r=-1 we have a perfect inverse correlation. In order to classify the intensity of the corre-183 lation between the independent and the dependent variable, we can consider the follow-184 ing ranges in absolute value: very weak association (r between 0 and 0.19), weak associa-185 tion (r between 0.20 and 0.39), moderate association (r between 0.40-0.59), strong associa-186 tion (r between 0.60 and 0.79), very strong association (r between 0.8 and 1).”
Reviewer 2 Report
Thank you for the opportunity to review this work exploring the correlation between the inflammatory profile and COVID-19 severity and mortality in neoplastic patients. The research question is interesting and the researchers have done extensive work. However, this paper has some major weaknesses, which I will outline in sections.
The abstract is too narrative, presents no data from the results and proposes conclusions that therefore seem unsupported by the evidence.
The introduction is very confusing, no logical discourse is followed to lead to an understanding of the rationale of the study. COVID-19 and the inflammatory state are mentioned, but it seems to be the one related to the development of COVID-19. Then it goes on to talk about the inflammatory profile in neoplastic patients with an excursus that is too long and didactic for an article. On the other hand, the topic of the relationship between a previous inflammatory state and the development of COVID-19 is not developed at all, and why precisely neoplastic patients were chosen for the study. Finally, it is inappropriate to anticipate the findings of the study at the end of the introduction.
There is a lack of information in the materials and methods about the rationale for what was collected, e.g. regarding laboratory values (considered for the whole stay?) and what was left out, e.g. length of stay, whether patients had already been admitted elsewhere and then transferred to pulmonology, etc. I find it very difficult to justify the methodology of correlating inflammatory marker values with the neoplastic state, whereas they may already be due to the onset of COVID-19 itself. The authors need to justify this better.
Results: The fact that the different malignancies are listed is not informative of the status of the patients (e.g. stage? - I ackowledge this is addressed in the limitations anyhow). Explanations of statistical methodology must be removed from the results. It is recommended to group the figures more to improve the readability of the results.
In the discussion, remarks are made about the relationship between severity of the inflammatory state and severity of COVID-19 disease, but in general it is never made clear what the specific correlation with the neoplastic state is, an element that also recurs in the conclusions. All didactic parts concerning markers should be removed from the discussion.
Diffused typos to be corrected and moderate revision of English required.
Moderate editing of English language required.
Round 2
Reviewer 2 Report
The authors have rearranged the paper substantially.
Minor changes can be suggested by the Assistant Editor in the final editing phase.